# Improving Screening Programmes for Sickle Cell Disorders and Other Haemoglobinopathies in Europe: The Role of Patient Organisations

**DOI:** 10.3390/ijns5010012

**Published:** 2019-01-29

**Authors:** John James, Elizabeth Dormandy

**Affiliations:** Sickle Cell Society, London NW10 4UA, UK

**Keywords:** sickle cell disorder, patient organisations, patient representatives, service users, sickle cell and thalassaemia screening programme, health policy

## Abstract

This discussion paper has been written to show the unique contribution and added value that Patient Organisations can give to the development and improvement of newborn screening programmes for sickle cell disorder (SCD) and other haemoglobinopathies in Europe. As an example, the action of the Sickle Cell Society (SCS) in partnership with statutory organisations in the U.K., such as the National Health Service (NHS) Sickle Cell and Thalassaemia Screening Programme (NHS SCT SP), will be described.

## 1. Background

Sickle cell disorders and thalassaemias are severe genetic disorders impairing haemoglobin function and/or production of the red blood cells. Both result in a significant morbidity and an increased risk of mortality, starting in the first years of life. Sickle cell disorder (SCD) has become the most common genetic disorder in several countries in Europe, most notably in France and the U.K., with overall prevalences of 1/1836 and 1/2439 newborns, respectively, in 2016 [1,2]. In Europe, only France, the U.K., the Netherlands and Spain have national newborn screening programmes for SCD. In the U.K., newborn screening has been set up in England and Scotland, and Wales and Northern Ireland have by and large followed the policy set in England. All these national programmes offer universal screening with the exception of France where a targeted programme is offered only to at-risk couples. Belgium has two regional programmes (Brussels and Liège) and there are pilot programmes in Italy, Germany and Ireland. Following migration and demographic changes there are an increasing number of people at risk of haemoglobinopathies in Europe, particularly in Germany and Italy. It has been noted that extension of the newborn screening is badly needed [3]. Recently, a consensus statement and recommendation for screening programmes has been produced [4].

Newborn screening programmes, with early implementation of comprehensive follow-up and prevention of the major complications, have dramatically improved survival in children with SCD in the U.S., the U.K. and France [5,6,7].

In the U.K., the NHS SCT Screening Programme is a linked antenatal and newborn screening programme. It uses the Family Origin Questionnaire and blood tests to screen pregnant women (and the baby’s biological father, where relevant) to identify those at risk of having a baby with either one or two serious inherited blood disorders—SCD and thalassaemia major. It also screens all newborn babies for SCD, as part of the newborn blood spot programme. Antenatal screening aims to offer pregnant women and their families reproductive choice. Newborn screening aims to identify affected babies, so that they can enter care and receive appropriate treatment before they become unwell. This can improve not only the quality of life of babies but also that of their parents/family. In the five-year period of 2010–2015, 1317 babies with sickle cell disease were identified in England [8]. In 2016/2017 677,000 pregnant women were screened and 667,500 newborn babies were screened [2].

The introduction of the NHS Sickle Cell and Thalassaemia Screening Programme in England has been the major driving force for improvements both in awareness and in the quality of care for children with SCD and other major haemoglobinopathies [9].

## 2. Role of the Patient Organisations

As a patient organisation, the Sickle Cell Society (SCS) works closely with people living with sickle cell, their families, NHS bodies (commissioners and providers), Government, Pharma Industry and a range of other national stakeholders and voluntary sector organisations.

Since the inception of the Screening Programme in 2001, the SCS has worked in partnership with the NHS Sickle Cell and Thalassaemia Programme and has been flexible in dealing with organisational changes within the NHS. The focus of that work has consistently been outreach work. For example, to address ignorance and stigma about SCD, the SCS has engaged with communities less likely to access health information through usual NHS channels, particularly men. The SCS produced a DVD entitled The Family Legacy to educate and improve knowledge about SCD in African communities [10]. Literature for families at risk has been produced jointly by the SCS and the Screening Programme, resulting in information that meets the needs of families with SCD [11,12]. The SCS has also acted as a bridge between health services and service users in the development of a National Haemoglobinopathy Register, resulting in a register that has ownership by service users. This in part is evidenced by the continuing increase of people living with SCD being registered on the National Haemoglobinopathy Register. As well as outreach work, the SCS has worked with the NHS SCT SP to influence policy through an All Party Parliamentary Group, Chaired by Diane Abbott MP [13].

The SCS has also worked with the NHS SCT SP and other organisations in developing and monitoring standards for the care of children and adults with SCD [14]. This work showed that 99% of screen positive babies were referred to a designated healthcare professional by 8 weeks of age and 85% of screen positive babies are seen in specialist treatment centres by 3 months [8]. The monitoring of these standards was a joint project between the SCS, U.K. Thalassaemia Society (UKTS) and NHS SCT SP. This joint collaboration ensured that patients had a say in how their data was being used. Ethical approval for the data collection specifically mentioned how important the engagement of the voluntary sector was in granting ethical approval. It is not possible to determine if the collaboration has led to increased diagnoses.

Now that the Screening Programme is well established, the focus has been on improving the screening pathway for pregnant women and their families. In March 2015, the SCS, UKTS and NHS SCT SP set up a small group of parents of children with SCD and thalassaemia together with health professionals from disciplines including obstetric, genetics and midwifery. The work of the group focussed on understanding and identifying the causes for why women with these conditions are tested late. The SCS in partnership with UKTS were commissioned to do this work on behalf of the NHS SCT SP [15]. As a direct result of the work of SCS and U.K. Thalassaemia Society and the lessons learned from the experiences of parents, the Screening Programme was able to update its standards and guidelines and public and professional educational resources. For example, an improved service pathway for at-risk couples was put in place by the Screening Programme. One of the most important lessons from this work was to bust the myth that late testing is due to late presentation by women. The majority of women first presented in pregnancy at less than 10 weeks of gestation and already knew they were carriers of the sickle cell or thalassaemia gene [15]. Collaborative working with the SCS and UKTS has been beneficial both for service users and health care professionals—it provides service users with a stronger voice and reduces the need for healthcare professionals to work with different service user groups. Other collaborations between patient organisations have been established, such as the Thalassaemia International Federation (TIF). One advantage for the Sickle Cell Society and the UKTS is the ability to work with one screening programme.

The SCS as a patient-led organisation provides expertise derived from work with patients/families on peer support, research and development, advocacy, education and policy development, as well as an independent patient perspective. This assists the NHS SCTP in gaining a better understanding of the patient/family perspective. It also assists them in assessing screening policy developments and the potential impact on the experience of service users.

Underlying this partnership approach is a recognition by the SCS that SCD is an underserved condition in the U.K., Europe and beyond, when compared to like inherited conditions such as Cystic Fibrosis. Our partnership is also focussed on addressing those inequities and reducing health inequalities.

## 3. Issues and Challenges

The austerity and financial challenges that face the NHS in the U.K., such as rising demand and a growing elderly population, place an increased burden on patient organisations such as the SCS to mobilise, advocate and empower patients/families to educate and work even more closely with external organisations such as the NHS Sickle Cell and Screening Programme.

Over the past five years, the SCS has changed significantly, and it continues to evolve. This change and evolution has been positive both internally and with external stakeholders such as the NHS SCT Screening Programme. In particular, we have enhanced our credibility and built strong relationships with our external Screening Programme colleagues. This is important because patient organisations have to continually demonstrate their credibility and professionalism without compromising their representativeness. This, in part, is evidenced by the SCS’s ability to secure a two-year tender from the Screening Programme between April 2016 and July 2018 to continue targeted outreach work. The SCS and UKTS have since been successful in bidding for a two-year extension to this work. Lack of funding and resources is a constant challenge, so contracts awarded by the public sector that extend beyond one year are particularly valuable to patient organisations.

## 4. Conclusions

The role played by the Sickle Cell Society as a Patient Organisation is constantly evolving in an ever-changing NHS landscape, which requires from us credibility, professionalism and the ability to deliver programmes of work supporting public sector organisations. Our partnership work with the Screening Programme is based on our core principles of representing the voice of SCD patients and families, but also based on openness, transparency, collaboration and equity.

Our outreach work with the Screening Programme over the past 18 years has not only helped raise awareness of the screening pathway for patients and health professionals but it has positively influenced the policies, guidelines and educational materials of the Screening Programme. This enables services that are better placed to meet service user needs and the programme objectives. It may be possible to extend this work by strengthening links between Patient Organisations in Europe.

The Sickle Cell Society believes strongly in working in partnership with statutory and non-statutory organisations as well as directly with people living with SCD and their families. This approach of partnership has delivered positive results for partners and service users in the U.K. We believe that there is potential to develop the role of Patient Organisations across Europe to work more collaboratively with the common purpose of improving health policy for people living with SCD in Europe. This approach is in place for other conditions such as thalassaemia through the Thalassamaemia International Federation (TIF). So why not for SCD? Given the rising prevalence of SCD in Europe and the inconsistent EU-wide models of care for SCD, collaborative working between Patient Organisations will become more important.

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
