# Peer review of "Improving Screening Programmes for Sickle Cell Disorders and Other Haemoglobinopathies in Europe: The Role of Patient Organisations"

_2409-515X, 2019, doi:10.3390/ijns5010012_

Round 1
Reviewer 1 Report
The authors of the paper show an excellent model of collaboration program. The bridge between institutional patterns and partnership as Sickle Cell Society is fundamental for the patient in order to be early in charge, followed up by health professionals and trained with educational materials. The knowledge and the awareness of one’s illness makes one's condition acceptable and promotes the recognition of symptoms. The implemented program allows to reduce the costs for National Health system. I hope that the English collaborative network could be a model for other European countries.
I have some minor comments:
1. lines 65-68: please reformulate the sentence to be clearer
2. lines 102-103: repetition “the change”?........“That change and evolution and change has been positive……”
3. Has the collaboration program produced an increase in the number of diagnoses?
4. How did the Sickle Cell Society organize the transition phase?
5. Please mention other publications, for example: Newborn screening for sickle cell disease in Europe: recommendations from a Pan-European Consensus Conference. Lobitz S et al.. British Journal of Haematology, 2018, 183, 648–660
Author Response
1. lines 65-68: please reformulate the sentence to be clearer - see lines 67-69
2. lines 102-103: repetition “the change”?........“That change and evolution and change has been positive……” - - see line 108-109
3. Has the collaboration program produced an increase in the number of diagnoses? - - see line 73-74
4. How did the Sickle Cell Society organize the transition phase?- see line 53-54
5. Please mention other publications, for example: Newborn screening for sickle cell disease in Europe: recommendations from a Pan-European Consensus Conference. Lobitz S et al.. British Journal of Haematology, 2018, 183, 648–660 - see line 30.
Reviewer 2 Report
Comprehensive management programs for haemoglobinopathies are recommended by WHO for management of inherited diseases. These programs should be managed at the community level. Thus, patients organisations, other patients, health professionals, should be partners in order to reach the objectives of such programs. Different European countries and the The European community are engaged in program where the patients organisations are major actors .
In this publication, the authors describe the experience of a British organization : the sickle cell society or SCS
SCS has a high level of organisation . "It was formed by a group of patients, parents and health professionals who were all concerned about the lack of understanding and the inadequacy of treatment for people living with sickle cell disorders". Their website has a good design and is well documented mainly on screening topics but also on management guidelines.
The presence of doctors and scientific advisers is a guarantee for the quality of scientific and medical data.
The experience of SCS shows that a patient organisation can be very efficient in improving screening programmes for sickle cell and other haemoglobinopathies. The efficiency is probably related to the level and quality of this organisation and activities.
The references of the publication are very recent. The patients in the organisation are volunteers and helped for screening by using tools and circumstances that health professionnels cannot do.
In the discussion chapter, a comparison about the experience and efficiency of an other powerful and old patients organisation such as TIFF is missing. It is required to understand what is new with the SCS : thus this comparison must be presented and discussed.
An other question is about the place and the role of small patients organisations? This experience suggest that small patients organisations have to federate together and with health professionals and scientists to be efficiently implicated in haemoglobinopathies management at the screening level.
The title talks about patient organisation. The SCS is a mixed organisation with patients, health professionals and scientists. The title has to be modified
Author Response
In the discussion chapter, a comparison about the experience and efficiency of an other powerful and old patients organisation such as TIFF is missing. It is required to understand what is new with the SCS : thus this comparison must be presented and discussed. - see line 90-93
An other question is about the place and the role of small patients organisations? This experience suggest that small patients organisations have to federate together and with health professionals and scientists to be efficiently implicated in haemoglobinopathies management at the screening level.- we have described our collaboration with UKTS in lines 76-91. This is an example of smaller organisations working well together
The title talks about patient organisation. The SCS is a mixed organisation with patients, health professionals and scientists. The title has to be modified - the SCS is a patient organisation governed by patients and their families . We do not think it is appropriate to change the title.